# A Molecular Interpretation of the Dynamics of Diffusive Mass Transport of Water within a Glassy Polyetherimide

**DOI:** 10.3390/ijms22062908

**Published:** 2021-03-12

**Authors:** Andrea Correa, Antonio De Nicola, Giuseppe Scherillo, Valerio Loianno, Domenico Mallamace, Francesco Mallamace, Hiroshi Ito, Pellegrino Musto, Giuseppe Mensitieri

**Affiliations:** 1Department of Chemical Sciences, University of Naples Federico II, Via Cintia, Complesso Monte S. Angelo, 80126 Napoli, Italy; andrea.correa@unina.it; 2Graduate School of Organic Materials Science, Yamagata University, 4-3-16 Jonan, Yonezawa, Yamagata 992-8510, Japan; ihiroshi@yz.yamagata-u.ac.jp; 3Department of Chemical, Materials and Production Engineering, University of Naples Federico II, Piazzale Tecchio 80, 80125 Naples, Italy; gscheril@unina.it (G.S.); valerio.loianno@unina.it (V.L.); 4Departments of ChiBioFarAm and MIFT-Section of Industrial Chemistry, University of Messina, CASPE-INSTM, V.le F. Stagno d’Alcontres 31, 98166 Messina, Italy; mallamaced@unime.it; 5Department of Nuclear Science and Engineering, Massachusetts Institute of Technology, Cambridge, MA 02139, USA; francesco.mallamace@unime.it; 6Institute on Polymers, Composites and Biomaterials, National Research Council of Italy, via Campi Flegrei 34, 80078 Pozzuoli, Naples, Italy

**Keywords:** water, polyetherimide, hydrogen bonding, diffusion, molecular dynamics

## Abstract

The diffusion process of water molecules within a polyetherimide (PEI) glassy matrix has been analyzed by combining the experimental analysis of water sorption kinetics performed by FTIR spectroscopy with theoretical information gathered from Molecular Dynamics simulations and with the expression of water chemical potential provided by a non-equilibrium lattice fluid model able to describe the thermodynamics of glassy polymers. This approach allowed us to construct a convincing description of the diffusion mechanism of water in PEI providing molecular details of the process related to the effects of the cross- and self-hydrogen bonding established in the system on the dynamics of water mass transport.

## 1. Introduction

Transport of water in polymeric systems is accompanied by hydrogen bonding self-interaction between water molecules and, frequently, by cross-interactions between water molecules and proton acceptor and proton donor groups present in the polymer backbone as well as self-interactions involving macromolecules. Interactional issues are relevant in a series of technological applications of polymers, as is the case of membranes for separation of gaseous and vapor mixtures, polymeric films with barrier properties to water vapor, environmental durability of polymer matrices for composites and humidity sensor applications [1,2,3,4,5,6]. Prompted by this motivation, several studies have been carried out to address the fundamental issue of understanding sorption thermodynamics of water in high performance glassy polymers. In particular, in a series of previous contributions by our group [7,8,9,10,11], the thermodynamics of polyimide–water systems was investigated, combining experimental approaches based on vibrational spectroscopy and gravimetric analysis with theoretical approaches based on Quantum Chemistry–Normal Coordinate Analysis (QC-NCA) and on an Equation of State (EoS) statistical thermodynamics theory based on a compressible lattice fluid model. To this aim, we adopted the Non-Random Hydrogen Bonding theory (NRHB), developed by Panayiotou et al. [12,13], that accounts for specific interactions as well as for non-random distribution of contacts between the lattice sites occupied by the components of the mixture and the empty sites. This theoretical framework, originally developed to address the case of sorption thermodynamics of low molecular weight compounds in rubbery polymers, has been then extended to deal with the case of glassy polymers [9]. To this purpose, the non-equilibrium nature of glassy systems was specifically taken into account by introducing the Non-Random Hydrogen Bonding–Non-equilibrium Theory for Glassy Polymers (NRHB-NETGP).

More recently [10], we have analyzed the sorption thermodynamics of a PEI–water system. To perform a comprehensive analysis of this interacting system, the information gathered from gravimetric and vibrational spectroscopy experimental investigations were combined not only with Quantum-Chemistry—Normal Coordinate Analysis (QC-NCA) and NRHB-NETGP theoretical approaches, but also by exploiting the wealth of information at the molecular level provided by Molecular Dynamics (MD) simulation. In fact, MD simulation delivered relevant evidence that was used to confirm and complete the physical picture emerging from the outcomes of vibrational spectroscopy and of macroscopic thermodynamics modeling. The results of this multidisciplinary approach allowed us to determine a comprehensive physical picture of the hydrogen bonding which establish within the system. The outcomes of MD simulations and of gravimetric and spectroscopic experimental analyses were in good qualitative and quantitative agreement with the results of statistical thermodynamics modeling (NRHB-NETGP). Notably, the amount of the different types of self- and cross-interactions were determined as a function of total concentration of water.

The present contribution, starting from the relevant results obtained in the previous analysis focused on equilibrium thermodynamics of a PEI–water system, is addressed to the exploration of the dynamics of mass transport of water in glassy PEI. After a short background section summarizing the most relevant results emerging from the equilibrium analysis, the new results on the dynamics of transport of water molecules within PEI matrix and on lifetimes of the different types of hydrogen bonding are presented. A molecular insight into diffusion mechanisms of water in PEI is provided by MD simulations, determining theoretical values for water intra-diffusion coefficient in PEI in the limit of vanishingly small concentrations. These values were found to be consistent with values of mutual diffusivity determined from time-resolved FTIR spectroscopy, in the same limit of small water concentration. In addition, the time-dependent behavior of HB bonds is presented, focusing on the mean bond lifetime that is the most accessible property reflecting this kind of behavior.

### 1.1. Background

#### 1.1.1. Relevant Results on Equilibrium Thermodynamics of a PEI–Water System

Vibrational spectroscopy provided the molecular level information onto which the NRHB-NETGP thermodynamic modeling is rooted. In particular, we considered the normal modes of the water molecule in the ν(OH) frequency range (3800–3200 cm^−1^), which were isolated by Difference Spectroscopy (DS), upon elimination of the polymer matrix interference [11]. Using this approach, it was also possible to determine the evolution with sorption time of the ν(OH) profile. The complex, partially resolved pattern, suggesting the occurrence of more than one species of penetrant, was interpreted with the aid of two-dimensional correlation analysis [10]. It was concluded that two couples of signals are present, each belonging to a distinct water species. In particular, the sharp peaks at 3655–3562 cm^−1^ were assigned to isolated water molecules interacting via H-bonding with the PEI backbone (cross-associated or first shell water molecules). The first shell adsorbate was found to have a 2:1 stoichiometry, with a single water molecule bridging two carbonyls (i.e., –C=O∙∙∙H–O–H∙∙∙O=C–). A second doublet at 3611–3486 cm^−1^ was associated with water molecules self-interacting with the first shell species through a single H-bonding (self-associated or second-shell water molecules). Analysis of the substrate spectrum revealed that the active sites (proton acceptors) on the polymer backbone are the imide carbonyls, while the involvement of the ether oxygens is negligible, if any.

A schematic diagram representing the two water species identified is reported in Figure 1.

Analysis of the ν(OH) band profile by least-squares curve fitting [10] allowed us to quantify the *ss* and *fs* population. Water species concentrations within the PEI, as determined at sorption equilibrium with water vapor at different pressures and at *T* = 303.15 K, in units relevant to the thermodynamic analysis, are represented in Figure 2 as a function of the content of water absorbed in the polymer. In agreement with FTIR, analysis MD simulations [10] identified two main different water populations: first shell, *fs*, and second shell, *ss*, water molecules. *Fs* water molecules interact directly with PEI carbonyl groups, while the *ss* water population consists of water molecules interacting with *fs* water molecules. The results of MD simulations highlighted how the *fs* water population mainly consists of water molecules bridging two consecutive intrachain carbonyls of the same PEI chain. Some interchain water bridges were also identified but they are reported to be present in a fraction from 0 to around 0.3 of all bridged water molecules, going from the lower water concentration to the higher water concentration system. Moreover, no significant involvement of PEI ether groups in hydrogen bond formation emerged from the MD results reported in ref [10].

In the same contribution, the thermodynamics of the PEI/water system at sorption equilibrium with a water vapor phase at prescribed pressure values has been analyzed on the basis of the NRHB-NETGP model for mixtures [10]. As anticipated, the NRHB-NETGP approach is a lattice fluid theory able to account for non-equilibrium nature of glassy polymers, for the presence of self- and cross-hydrogen bonding and for non-random mixing of the two components. The reader is referred to the relevant literature [9,11] for the relevant equations of the -NRHB-NETGP model. It suffices here to remind that the application of this theory provides a quantitative prediction on the type and number of hydrogen bonds formed at equilibrium within the system, once that the model has been used to fit experimental sorption isotherms of a penetrant within a polymer. In particular, the results of the application of NRHB-NETGP theory to the PEI/H_2_O system, evidenced that the predictions on HB formation at equilibrium agree very well with the experimental results obtained by in situ infrared spectroscopy and with the theoretical results obtained by MD simulations [10]. This is evident in Figure 2, where this comparison is reported with reference to the HB interactions actually established in the system, i.e., water self-interactions (indicated by “11” subscript) and water–PEI (carbonyl group) cross-interactions (indicated by “12” subscript) as a function of mass fraction of water, ω_1_. In Figure 2 n_11_/m_2_ and n_12_/m_2_ stand for the mmol of H-bond formed per gram of polymer.

**Figure 2 ijms-22-02908-f002:**
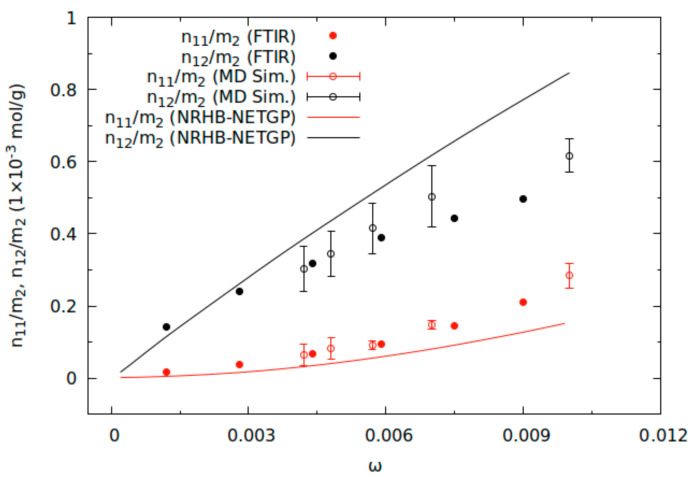
Comparison of the predictions of the NRHB-NETGP model for the amount of self and cross-HBs with the outcomes of FTIR spectroscopy and of MD simulations. Reprinted with permission from the authors of [10]. Copyright (2017) American Chemical Society.

On this basis, it was concluded that the NRHB-NETGP theory provides a reliable expression of the chemical potential of H_2_O in glassy PEI. This expression will be used in the present contribution to evaluate the thermodynamic factor that appears in the theoretical expression of PEI/H_2_O mutual diffusivity, as detailed in the following section.

#### 1.1.2. Mutual vs. Intra-Diffusion Coefficients

Diffusive mass transport of small molecules in polymers is a multifaceted phenomenon whose description, in the most complex cases, involve concurrently mass, momentum, and energy balances with the introduction of thermodynamically consistent constitutive equations for the different types of flux implicated and for the relevant material properties. As a starting point, it is important to provide a description of the basic approaches used to express the mass flux of a component *i* in a mixture. We will address here the general case, although our final goal is to deal with mass transport in an isotropic system formed by a low molecular weight compound (penetrant) dissolved within a polymer matrix.

In a binary mixture, the total mass flux of component *i*, ${\underline{n}}_{i}$, referred to a lab fixed frame of reference is expressed as in Equation (1):(1)$${\underline{n}}_{i}=\rho_{i}{\underline{u}}_{i}={\underline{j}}_{i}^{M}+\rho_{i}{\underline{u}}^{M}$$
or, equivalently, as in Equation (2):(2)$${\underline{n}}_{i}=\rho_{i}{\underline{u}}_{i}={\underline{j}}_{i}^{V}+\rho_{i}{\underline{u}}^{V}$$
where, as in Equation (3),
(3)$${\underline{u}}^{M}\equiv \omega_{1}{\underline{u}}_{1}+\omega_{2}{\underline{u}}_{2}$$
is the mass average mixture velocity referred to a lab fixed frame of reference while, in Equation (4)
(4)$${\underline{u}}^{V}\equiv c_{1}\bar{\upsilon_{1}}{\underline{u}}_{1}+c_{2}\bar{\upsilon_{2}}{\underline{u}}_{2}$$
is the volume average mixture velocity referred, again, to a lab fixed frame of reference. In the previous equations *c_i_*, $\rho_{i}$, $\omega_{i}$, and $\bar{\upsilon_{i}}$ represent, respectively, the molar concentration of component *i*, the mass of component *i* per volume of mixture, the mass fraction of component *i*, and the partial molar volume of component *i*. The symbol ${\underline{u}}_{i}$ represents the velocity of molecules of component *i*, referred to a lab fixed frame of reference. In the present context, we deal with the specific case of penetrant–polymer mixtures and we will refer to the penetrant with subscript 1 and to the polymer with the subscript 2. From the previous Equations (1) and (2) it is readily derived that, as in Equations (5) and (6),
(5)$${\underline{j}}_{i}^{M}\equiv \rho_{i}\cdot ({\underline{u}}_{i}-{\underline{u}}^{M})$$
(6)$${\underline{j}}_{i}^{V}\equiv c_{i}\cdot ({\underline{u}}_{i}-{\underline{u}}^{V})$$

Equations (5) and (6) define, respectively, the diffusive mass flux of component *i* relative to the weight average velocity of the mixture, ${\underline{j}}_{i}^{M}$, and the diffusive molar flux of component *i* relative to the volume average velocity of the mixture, ${\underline{j}}_{i}^{V}$. If the contributions to mass flux determined by a gradient of temperature, a gradient of pressure and by the difference of the body forces acting on unit of mass of each component can be neglected, the following constitutive Equation (7) holds for ${\underline{j}}_{i}^{M}$ [14]:(7)$${\underline{j}}_{i}^{M}=-D_{12}\rho \underline{\nabla }\omega_{i}$$
where *ρ* is the density of the mixture. In the case of constant density, Equation (7) takes the classical form of the so-called Fick’s first law [14] in Equation (8):(8)$${\underline{j}}_{i}^{M}=-D_{12}\underline{\nabla }\rho_{i}$$
where *D*_12_ is the mutual diffusivity of the “12” system. Note that, as (see Equation (9))
(9)$$\sum_{i}{\underline{j}}_{i}^{M}=1$$
and Equation (10),
(10)$$\underline{\nabla }\omega_{1}=-\underline{\nabla }\omega_{2}$$
a single mutual-diffusion coefficient *D*_12_, is defined intrinsically by Equation (7) for both components. In fact, as we will see in the following, this coefficient is a property of the binary system and is a function of temperature and concentration.

Considering the specific case of isothermal diffusion of water in an unconstrained film of PEI, it is noted that, at the investigated conditions (*T* = 303.15 K, range of relative pressure of water vapor, *p*/*p*_0_ = 0/0.6), the weight fraction of water within the polymer is always lower than 0.01. This implies that no relevant stresses develop as consequence of water sorption. The low amount of penetrant absorbed combined with the absence of polymer swelling allows also the assumption of a constant mixture density. In addition, the bulk velocity of the polymer/water mixture can be considered to be negligible (i.e., ${\underline{u}}^{M}≅0;\;{\underline{u}}^{V}≅0$), in view of the low intrinsic mobility of polymer (i.e., ${\underline{u}}_{2}≅0$), that is the largely prevailing component. Moreover, Equation (7) can be taken as a constitutive expression of the diffusive mass flux as, in the case at hand, other driving forces beside the composition gradient can be ruled out. In fact, (i) the driving force related to the difference of the body forces acting per unit of mass of each component is equal to zero since in this case they are only associated to the gravitational field, (ii) the driving force related to the gradient of temperature is zero in view of the isotheral condition, and (iii) the driving force related to the gradient of pressure is zero in view of the uniformity of the state of stress. Finally, note that the transport of fluids in polymers is slow enough to assure that also the inertial contributions can be neglected. Therefore, as in Equation (11),
(11)$${\underline{n}}_{i}≅{\underline{j}}_{i}^{M}≅{\underline{j}}_{i}^{M}=-D_{12}\underline{\nabla }\rho_{i}$$

In such a case the 1-*D* differential mass balance on component “*i*” reads [14], as in Equation (12),
(12)$$\frac{\partial \rho_{i}}{\partial t}=-\frac{\partial n_{i}}{\partial x}=-\frac{\partial }{\partial x}\left(-D_{12}\frac{\partial \rho_{i}}{\partial x}\right)$$
where *x* is a lab fixed coordinate. Equation (12), in the case of *D*_12_ independent of composition, takes the form of the so-called Fick’s second law [15], as in Equation (13):(13)$$\frac{\partial \rho_{i}}{\partial t}=D_{12}\frac{\partial^{2}\rho_{i}}{\partial x^{2}}$$

Under the same hypotheses, and based upon a well-established statistical mechanics framework, Bearman [16] developed a constitutive equation for ${\underline{j}}_{i}^{V}$ in case of a mono-dimensional binary diffusive problem. In this approach it is assumed that, after a chemical potential gradient is established within the binary mixture (due to a concentration gradient), the system attains a local quasi-stationary regime in which the driving force to the diffusion mechanism of a molecule of component *i* (given by its chemical potential gradient) is mechanically balanced by a frictional force deriving from intermolecular interactions (consisting of both self-interactions, *i-i*, and cross interactions, *i-j* with j≠i ). Bearman derived an expression for the frictional forces involving the definition of friction coefficients $\zeta_{ij}$ (i,j = 1.2) which obey to a reciprocal relationship (i.e.,$\zeta_{ij}=\zeta_{ji}$), and, based on this approach, he has also defined a mutual-diffusion coefficient, $D_{12}^{V}$, analogous to the mass diffusional coefficient $D_{12}$, such that [16], as in Equation (14),
(14)$$j_{i}^{V}=-D_{12}^{V}\frac{dc_{i}}{dx}$$
where, as in Equation (15),
(15)$$D_{12}^{V}\equiv \frac{\bar{\upsilon_{1}}}{\zeta_{12}}RT\left[1+{\left(\frac{\partial \ln(f_{2})}{\partial \ln\left(c_{2}\right)}\right)}_{T,P}\right]=\frac{\bar{\upsilon_{2}}}{\zeta_{12}}RT\left[1+{\left(\frac{\partial \ln(f_{1})}{\partial \ln\left(c_{1}\right)}\right)}_{T,P}\right]\equiv \;D_{21}^{V}$$

In Equation (15), *f_i_* represents the activity coefficient of component *i* in the binary system. The equality of the two mutual volumetric diffusion coefficients appearing in Equations (14) and (15), follows from the Gibbs–Duhem equation and from the definition of volume average velocity.

In view of the simplifying assumptions discussed before, legitimated by the low value of penetrant concentration, it can be derived a relationship involving $D_{12}^{V}$ and $D_{12}$ for the generic component *i* in the case of mono-dimensional diffusion taking place in direction *x*, as in Equation (16),
(16)$$M_{i}D_{12}^{V}\frac{dc_{i}}{dx}=D_{12}^{V}\frac{d\rho_{i}}{dx}≅D_{12}^{}\frac{d\rho_{i}}{dx}\;$$
from which one obtains Equation (17):(17)$$D_{12}^{V}\approx D_{12}$$
where *M_i_* represents the molecular molar weight of component *i*.

It is useful to introduce now the so-called intra-diffusion coefficients [17] that represent the intrinsic diffusive mobility of each component in a binary mixture, i.e., in the absence of any driving force for the mass flux (e.g., gradients of chemical potential, temperature, pressure). These coefficients are indicated as, respectively, *D*_1_ and *D*_2_. Expressions have been proposed relating the mutual diffusivity in a binary mixture, *D*_12_, or, similarly, any other type of mutual-diffusion coefficient referred to a different frame of reference, to the intra-diffusion coefficients of the two components.

On the grounds of statistical mechanics, the three kinds of diffusion coefficients—*D*_12_, *D*_1_, and *D*_2_—can be expressed in terms of the molecular friction coefficients (in the case at hand, penetrant–penetrant, polymer–polymer, and penetrant–polymer friction coefficients, respectively, denoted by *ζ*_11_, *ζ*_22_, and *ζ*_12_) [16], as in Equations (18)–(20).
(18)$$D=\frac{M_{2}\cdot \omega_{1}\cdot {\hat{V}}_{2}}{N_{A}^{2}\cdot \zeta_{12}}{\left(\frac{\partial \mu_{1}}{\partial \omega_{1}}\right)}_{T,\;P}$$
(19)$$D_{1}=\frac{RT}{N_{A}^{2}\left(\frac{\omega_{1}\cdot \zeta_{11}}{M_{1}}+\frac{\omega_{2}\cdot \zeta_{12}}{M_{2}}\right)}$$
(20)$$D_{2}=\frac{RT}{N_{A}^{2}\left(\frac{\omega_{2}\cdot \zeta_{22}}{M_{2}}+\frac{\omega_{1}\cdot \zeta_{12}}{M_{1}}\right)}$$
where *N_A_* is the Avogadro’s number.

Actually, free volume theories provide independent expressions (see, for example, in [18,19]) for intra-diffusion coefficients, *D*_1_ and *D*_2_, that do not need the knowledge of friction coefficients. However, since three friction coefficients appear in Equations (18)–(20) in general, it is not possible to express *D*_12_, only in terms of *D*_1_ and *D*_2_ (i.e., with no friction coefficients). This is, however, possible if special circumstances occur, e.g., if one is able to write a relationship linking the three friction coefficients, or if one considers the limit of trace amount of penetrant, or in the cases where *D*_1_ > *D*_2_. For instance, assuming that *ζ*_12_ is the geometric mean of *ζ*_11_ and *ζ*_22_ [18,19] or, alternatively, assuming that the ratio between the friction coefficients is constant [16] it is possible to obtain the following relationship, as in Equation (21):(21)$$D_{12}≅D_{12}^{V}=\frac{\left(D_{2}x_{1}+D_{1}x_{2}\right)}{RT}{\left(\frac{\partial \mu_{1}}{\partial \ln x_{1}}\right)}_{T,P}=\frac{\left(D_{2}x_{1}+D_{1}x_{2}\right)}{RT}{\left(\frac{\partial \mu_{2}}{\partial \ln x_{2}}\right)}_{T,P}$$
where $\mu_{i}$ and *x_i_* represent, respectively, the molar chemical potential and the molar fraction of component *i*, and *P* and *T* represent, respectively, the spatial uniform pressure and temperature of the binary mixture. In the present context *D*_1_ > *D*_2_ and the water molar fraction range is approximately 0.94–0.97, thus assuring that it is also $D_{1}x_{2}>D_{2}x_{1}$. Therefore, the relationship (21) reduces to an explicit relationship relating the measured mutual-diffusion coefficient $D_{12}$ just, to the intra-diffusion coefficient of water *D*_1_, as in Equation (22):(22)$$D_{12}^{}≅\frac{D_{1}}{RT}{\left(\frac{\partial \mu_{1}}{\partial \ln x_{1}}\right)}_{T,P}$$

In order to estimate, exclusively on a theoretical basis, the value of *D*_12_ from Equation (22), one then needs to know the expressions of *D*_1_ and *μ*_1_. In the present investigation, the value of the intra-diffusion coefficient has been retrieved from MD simulations of a PEI/H_2_O system with uniform concentration, by averaging the statistics of the evolution of the diffusion path with time of each single water molecule. The estimate of *μ*_1_ as a function of concentration has been instead obtained by using the NRHB-NETGP thermodynamic model. The parameters of this for the water/PEI system model are available in a previous publication by our group [10]. The set of equations involved in the calculation of *μ*_1_ according to the NRHB-NETGP must be solved numerically, so that only an implicit expression for the penetrant molar chemical potential as a function of concentration at a given pressure and temperature is available. Therefore, the derivative of the NRHB-NETGP penetrant molar chemical potential appearing in Equation (22) has been evaluated numerically. In particular, it has been estimated assuming a centered difference finite scheme with a variable concentration step equal to $10^{-6}c_{1}$. This step has provided an excellent compromise between the accuracy of the approximated numerical scheme adopted and the round-off error deriving from the finite digit arithmetic associated to the calculator used.

The estimates of mutual diffusivity, *D*_12_, obtained from Equation (22) for the PEI/H_2_O system, based on information provided by MD calculation for *D*_1_ and by NRHB-NETGP model of mixture thermodynamics for *μ*_1_, will be compared with the experimental values obtained independently by in situ time-resolved infrared spectroscopy.

Finally, note that in the limit of a vanishingly small mass fraction of penetrant (water) in the penetrant–polymer systems—and thus in the limit of vanishingly small relative pressure of penetrant (water) vapor—mutual-diffusion coefficient, *D*_12_, and intra-diffusion coefficient of the penetrant, *D*_1_, converge to the same value [18].

## 2. Results and Discussion

### 2.1. Determination of Mutual Diffusivity of the Water–PEI System from Vibrational Spectroscopy

FTIR spectroscopy has been shown to be a powerful tool to investigate water diffusion in polyimides [11]. For the case at hand, the process can be suitably monitored by considering the normal modes of the diffusing molecule in the ν(OH) frequency range (3800–3200 cm^−1^). In fact, difference spectra can be collected in this range as a function of time, providing an accurate evaluation of the sorption/desorption kinetics.

The experimental data were analyzed in terms of the PDE expressing the Fick’s second law of diffusion introduced in the background section (see Equation (13)). For the case of a plane sheet exposed to symmetric boundary conditions (i.e., an equal penetrant activity on both sides), the solution of Equation (13) can be expressed as [15,20]
(23)$$\frac{A\left(t\right)}{A\left(\infty \right)}=\frac{M\left(t\right)}{M\left(\infty \right)}=1-\frac{8}{\pi^{2}}\sum_{m=0}^{\infty }\frac{1}{{\left(2m+1\right)}^{2}}\cdot \exp\left[\frac{D_{12}{\left(2m+1\right)}^{2}\pi^{2}t}{L^{2}}\right]$$
where *M(t)* and *M(∞)* represent, respectively, the total mass of penetrant absorbed in the polymer sheet at time *t* and at equilibrium, while *A(t)* and *A(∞)* represent, respectively, the absorbance area of the analytical band at time *t* and at equilibrium (integration limits 3800–3250 cm^−1^) and *L* is the sample thickness. Equation (23) has been used to best fit experimental sorption kinetics data using *D*_12_ as fitting parameter, assumed to be independent on concentration.

In Figure 3, the experimental water sorption kinetics for a step increase of relative pressure of water vapor from 0 to 0.6 (“integral sorption test”) is reported. The inset displays the ν(OH) water band at increasing sorption times. The normalized absorbance of the analytical band, *A(t)/A(∞)*, is plotted as a function of the square root of time (Fick’s diagram) for an experiment performed at 303.15 K. The very good fit of the experimental data and the linear dependence of the *A(t)/A(∞)* on the square root of time for ordinate values up to around 0.6, point to the so-called, Fickian behavior of the system [15]. The water–PEI mutual diffusivity was found to be 1.52 × 10^−8^ cm^2^/s, that is in good agreement with previous literature reports on commercial polyimides [21].

The kinetic analysis of the diffusion process was also performed in the p/p_0_ interval from 0 to 0.6 performing “differential sorption tests”, i.e., increasing stepwise by a 0.1 increment the relative pressure of H_2_O vapor. The related Fick’s diagrams are reported in Figure 4. Consistently with the assumption of a constant diffusivity, the *D*_12_ values obtained from the fitting of kinetics data using Equation (23) are rather independent of concentration (average value: *D*_12_ = 1.55 × 10^−8^ ± 0.03 × 10^−8^ cm^2^/s); only at *p*/*p*_0_ = 0.1 the diffusivity is appreciably lower (*D*_12_ = 1.37 × 10^−8^± 0.03 × 10^−8^ cm^2^/s).

The values of water–PEI mutual diffusivity coefficients as determined by best fitting the experimental sorption kinetics data using Equation (23) are collectively reported in Figure 5.

### 2.2. Molecular Dynamics Simulations

Before performing simulations of water diffusion within the PEI/water system, relaxation of PEI atomistic model has been obtained to generate a well-equilibrated system of full atomistic polymer melts at 570 K, followed by fast quenching from 570 to 303.15 K, using a MD-SCF approach, as reported in [10]. The obtained configurations, corresponding to “System I” reported in Table 1, have been used to build up systems at different water concentrations by insertion of water molecules.

Values of intra-diffusion coefficient of water have been theoretically calculated on the basis of the mean square displacement as a function of time of each water molecule present within a PEI domain resulting from molecular dynamics simulations at several uniform water concentrations. The values of intra-diffusion coefficient of water as determined from the simulations performed at *T* = 303.15 K are reported in Figure 6 as a function of mass fraction of water.

**Table 1 ijms-22-02908-t001:** Systems composition and simulation details.

System	Box (nm^3^)	ω	Water Molecules	Total Particles	Simulation Time (ns)
I	6.31000	-	0	22,680	120
II	6.31308	0.0042	86	22,938	198
III	6.31358	0.0048	100	22,980	200
IV	6.31589	0.0057	120	23,040	200
V	6.31980	0.007	150	23,130	200
VI	6.32867	0.01	220	23,340	240

As anticipated, the intra-diffusion coefficient represents the absolute intrinsic mobility of a water molecule within the PEI/water mixture in the absence of any gradient of water chemical potential and of any other driving force for mass transport. Conversely, the mutual-diffusion coefficient represents water mobility as referred to the mass average velocity of the polymer–water mixture, under the action of a gradient of chemical potential of water and/or of other driving forces. In general, these two coefficients have different values. However, based on reasonable assumptions, it has been already discussed that, in the limit of vanishingly small mass fraction of water in the water–polymer systems—and thus in the limit of vanishingly small relative pressure of water vapor—mutual diffusion coefficient and intra-diffusion coefficient tend to the same value. Actually, the diffusivity value estimated from FTIR spectroscopy measurements and the intra-diffusion coefficient predicted on the basis of MD simulation apparently converge to a common value of about 1.20 × 10^−8^ cm^2^/s at a vanishingly small water concentration, thus confirming the consistency of MD simulations.

In order to have a qualitative molecular interpretation of the possible molecular nature of different dynamic states of water molecules, as indicated by the FTIR spectra analysis, the behavior of intra-diffusion coefficients obtained on the basis of MD simulations, averaging over all simulated water molecules, is decomposed as a distribution. In particular, the distribution of the diffusion coefficient obtained from each water molecule as a histogram for two different water concentrations is reported in Figure 7A,B. From the Figure 7A, it is clear that a tail of faster diffusing water molecules is obtained for the system at higher water concentration as compared to the system at lower concentration (Figure 7B).

This behavior can be interpreted in the light of the information collected in a previous contribution [10] about the state of water molecules, at equilibrium, within the PEI/water mixture as a function of their concentration. In fact, at low concentration, water molecules are prevalently present as first shell water. First shell water is mainly contributed by molecules bridging, by hydrogen bonds, two consecutive carbonyl groups present along a macromolecule and, at a lesser extent, by molecules bridging, by hydrogen bonds, two non-consecutive carbonyls located on two different macromolecules, or to different repeating units of the same macromolecule. Conversely, as the water concentration increases, the concentration of so-called second shell water molecules increases. Second shell water refers to those water molecules that interact, by a single hydrogen bond, with a first shell water molecule. Due to the structure of the interaction complex, first shell water is characterized by a stronger energy of interaction with the PEI carbonyls as compared with the energy of interaction of second shell water molecules with a first shell water molecule.

It is then expected that at low water concentration a lower mobility (diffusivity) should be observed and that mobility should increase with concentration, in agreement with reported results of MD simulations. In order to deepen understanding of this effect, water molecules have been grouped in sets according to their diffusion coefficient and the fraction of the simulation time spent in the first shell state has been calculated averaging over each molecule set. The results of this analysis are reported in Figure 7C. From this figure it is clear that sets of water molecules spending a large fraction of simulation time in the first shell bridging state are characterized by a lower mobility. On the contrary, larger diffusion coefficients are obtained for sets of water molecules spending most of the simulation time in states different from first shell.

### 2.3. Comparison of Theoretical Predictions with Results of Vibrational Spectroscopy

In Figure 8, a comparison is reported between the values of the mutual-diffusion coefficient, *D*_12_, determined experimentally by FTIR spectroscopy and discussed in Section 2.1, and the values of this coefficient predicted using in Equation (22) the values of *D*_1_ estimated by MD calculations and the values of ${\left(\frac{\partial \mu_{1}}{\partial \ln x_{1}}\right)}_{T,P}$ estimated using the NRHB-NETGP model. In order to compare experimental results with theoretical findings, values of mutual diffusivity estimated from the experimental differential sorption steps by fitting sorption kinetics using Equation (23) are reported as a function of the average water mass fraction present within the polymer during the test, calculated as the arithmetic average of the uniform initial and final water mass fraction.

As already discussed in Section 2.1, the experimental results obtained by FTIR spectroscopy point out that, in the whole range investigated, the mutual diffusion coefficient is roughly constant as a function of penetrant concentration. The theoretical values of *D*_12_ seemingly approach the experimental values when water concentration tends to zero. We remind that, in this limit, the theoretical values of *D*_12_ and *D*_1_ tend to the same value. Conversely, as the concentration increases, a gradually increasing departure of the theoretical values of *D*_12_ from the experimental values is evident. This mismatch could be attributed to the fact that, as reported in literature [22], the MD approach implemented here is reliable at quite low penetrant concentration while it provides a progressively increasing overestimation of the dependence of intra-diffusion coefficient as the water concentration increases and, in turn, an increasing overestimation of *D*_12_ values.

### 2.4. H-Bond Lifetimes

An interesting additional information that can be obtained from MD simulations is the time-dependent behavior of the H-bonds, the most accessible property reflecting this kind of behavior being the mean bond lifetime.

In order to estimate H-bond lifetimes, we extracted from the simulation data the time-dependent autocorrelation functions of state variables which reflect the existence (or non-existence) of bonds between each of the possible donor acceptor pairs. In accordance with the works in [23,24,25,26,27], the HB correlation function *C*(*t*) is defined as:(24)$$C\left(t\right)=\underset{\mathrm{ij}}{\sum }s_{\mathrm{ij}}\left(t_{0}\right)s_{\mathrm{ij}}\left(t_{0}+t\right)/\underset{\mathrm{ij}}{\sum }s_{\mathrm{ij}}\left(t_{0}\right)$$
where the dynamical variable *s_ij_*(*t*) equals unity if the particular tagged pair of molecules is hydrogen bonded and is zero otherwise. The sums are over all pairs and *t*_0_ is the time at which the measurement period starts (C(0) =1).

The H-bond lifetime, τ, is readily defined from the exponential decay of *C*(*t*):(25)$$C\left(t\right)=A\exp\left(-t/\tau \right)$$

In our analysis, *C(t)* of H-bond functions has been calculated as continuous hydrogen bond correlation functions, meaning that each *s_ij_* variable is allowed to make just one transition from unity to zero when the H-bond is first observed to break, but is not allowed to return to unity should the same bond reform subsequently. On the basis of the results reported in [10], we confined our analysis to one kind of HB acceptor that is the carboxylic group (defined as AC1 in [10]). In Figure 9, *C(t)* corresponding to system II (the one containing 86 water molecules) and system VII (the one containing 220 water molecules) is reported. Interestingly, no reasonable fitting of *C(t)* correlation functions was obtained using a single exponential decay (see dashed lines in both figures). Instead, a better agreement has been obtained using a sum of two different exponential decays, i.e., in Equation (26):(26)$$C\left(t\right)=A_{1}\exp\left(-t/\tau_{1}\right)+A_{2}\exp\left(-t/\tau_{2}\right)$$

We identify a fast decay of about 4 ps and a slower one going from about 2 ns for the system at low water concentration to 118 ps for the system at higher water content, see Table 2. A similar range of lifetimes has been reported in other simulation analyses, also indicating values in excess of 1 ns for interacting glassy polymers [28]. Moreover, this feature is in agreement with the identification of two water populations: one consisting in water molecule bridging two carbonyls of PEI (slower decay) and one consisting of water molecules interacting with first shell water molecules (faster decay). In [10], as already recalled in the previous sections, we have also demonstrated that two types of first shell bridging HB can exist: intrachain and interchain first shell HB, depending on if the two carbonyls are consecutive on the same chain or belonging to separate chains.

Interchain water bridges are present at lower extent going from a fraction of 0, at the lowest water concentration (system II), to a fraction around 0.3, of all bridged molecules, at higher water concentration (system VII). Structural analysis clearly shows that in the case of first shell intrachain HB the distances between the carbonyl oxygens and water’s oxygens are shored and their distribution are narrow, compared to those of first shell interchain hydrogen bonded water molecules, indicating higher mobility of the latter with respect to the former ones. Therefore, the increase in interchain first shell HB in a higher water concentration system further contributes to decrease H-bond lifetime for the first shell HB population.

As final remark, it is worth noticing that lifetimes of the order of several picoseconds for the second shell water molecules are consistent with the experimental observation of two distinct signals generated by this species in the vibrational spectrum. In fact, as the characteristic decay time of vibrational transitions is of the order of a picosecond, shorter H-bonding lifetimes would produce a fully convoluted bandshape rather than the well-resolved profile that is observed.

**Table 2 ijms-22-02908-t002:** HB Lifetime values and relative population weights (A_1_ and A_2_) obtained fitting the HB autocorrelation functions by Equation (26).

System	*τ* _1_	*τ* _2_	A_1_	A_2_
II	4.4 ps	2362 ps	0.62	0.38
VI	3.9 ps	118 ps	0.67	0.33

## 3. Experimental

### 3.1. Materials

Amorphous PEI with ${\bar{M}}_{n}=1.2\;\times 10^{4}\;Da,\;{\bar{M}}_{w}=3.0\times \;10^{4}\;Da,\;T_{g}=210\;{}^{\circ}C,Density=1.260\;g/cm^{3}$ was kindly supplied by Goodfellow Co., Coraopolis, PA, USA, in the form of a 50.0 μm thick film. Film thicknesses suitable for FTIR spectroscopy were obtained by dissolving the original product in chloroform (15% *wt*/*wt* concentration), followed by solution casting on a tempered glass support. Film thickness in the range 10 to 40 μm was controlled by using a calibrated Gardner knife to spread the solution over the support. The cast film was dried 1 h at room temperature and 1 h at 80 °C to allow most of the solvent to evaporate, and at 120 °C under vacuum overnight. At the end of the drying protocol, the film was removed from the glass substrate by immersion in distilled water at 80 °C. Milli-Q water was used in all sorption experiments.

### 3.2. FTIR Spectroscopy

Time-resolved FTIR spectra of polymer films exposed to water vapor at a constant relative pressure (*p/p_0_*) were collected in the transmission mode, monitoring the characteristic signature of the penetrant up to the attainment of sorption equilibrium. The sorption experiments were performed in a custom designed, vacuum-tight cell positioned in the sample compartment of the spectrometer. This cell was connected through service lines, to a water reservoir, a turbo-molecular vacuum pump, and pressure transducers. Full details of the experimental setup are reported in [29]. Before each sorption measurement, the sample was dried under vacuum overnight at the test temperature in the same measuring apparatus. The FTIR spectrometer was a Spectrum 100 from PerkinElmer (Norwalk, CT, USA), equipped with a Ge/KBr beam splitter and a wide-band deuterated triglycine sulfate (DTGS) detector. Parameters for data collection were set as follows: resolution = 2 cm^−1^, optical path difference (OPD) velocity: 0.5 cm/s, and spectral range: 4000−600 cm^−1^. A single spectrum collection took 2.0 s to complete under the selected instrumental conditions. Continuous data acquisition was controlled by a dedicated software package for time-resolved spectroscopy (Timebase from PerkinElmer, Norwalk, CT, USA). The Absorbance spectrum of the penetrant was obtained by use of the single-beam spectrum of the cell containing the dry sample as background [10].

### 3.3. MD Simulations

#### 3.3.1. Polymer Model

The atomistic model of PEI used in the present work has be taken from in [10]. The model is based on the OPLS-AA force-field [30,31,32,33] from which bonded and non-bonded interaction parameters were taken. As shown in [10], the employed model can correctly reproduce the X-ray scattering pattern of the polymer bulk, the mass density PEI, and the relative amount of hydrogen bonds (compared with experimental FTIR measurements) in the system PEI/water. For non-bonded interactions, a cut-off of 1.1 nm was used. Coulomb interactions were treated by generalized reaction field [34] scheme with a dielectric constant ε = 5 and a cut-off of 1.1 nm. Chemical structure of PEI repeating unit is shown in Scheme 1. Each PEI chain used in this work contains 12 repeating units and each chain is terminated by a phenyl group and a hydrogen atom. In PEI/water systems, water molecules were described by the simple-point-charge (SPC) model [35]. Full details on the potentials used to treat non-bonded and bonded interactions and on the values of their parameters are reported in [10].

#### 3.3.2. Simulation Details

Hybrid particle-field molecular dynamics technique (MD-SCF) [36,37] was employed to equilibrate PEI pure amorphous. Simulation runs have been performed by OCCAM code [38] in the constant volume and temperature (*NVT*) ensemble, following the same procedure reported by De Nicola et al. [39], with the temperature fixed at 570 K, controlled by the Andersen thermostat a collision frequency of 7 ps^-−1^, a timestep of 1 fs, and density field density update performed every 0.1 ps. For more details see in [36,37,39].

GROMACS package [40] was employed for all atomistic MD simulations. Pure PEI systems were preliminarily equilibrated for 1 ns in NVT ensemble (starting from MD-SCF relaxed structures). All production runs were performed in the constant pressure and temperature (NPT) ensemble, by a timestep of 2 fs. Periodic boundary conditions were applied and, for all systems, a constant number of 27 PEI chains were considered. The temperature was fixed at 303.15 K by Berendsen thermostat (coupling time 0.1 ps). The pressure was kept constant at 1.01325 bar by Berendsen barostat (coupling time 0.1 ps) [41]. Table 1 reports composition and simulation details for all simulated systems. For systems I and II five independent MD simulations, with different starting configurations, were performed. For systems with higher water content (i.e., III, IV, V, and VI) two independent MD simulations, with different starting configurations, were performed.

## 4. Conclusions

The diffusion of water in PEI as determined experimentally by time-resolved FTIR spectroscopy has been interpreted on the basis of MD simulations combined with modeling of water chemical potentials by means of a non-equilibrium lattice fluid model for thermodynamics of water/PEI system. Based on the physical picture on H-bonding formation obtained in a previous investigation, the diffusion process has been investigated by MD simulations of systems with different compositions. The results of the theoretical analysis are quantitatively consistent with the experimental results provided by FTIR spectroscopy, in terms of mutual diffusion coefficient, in the limit of vanishingly small water concentration. However, the predictions obtained from the theoretical analysis provide values of mutual diffusivity that increasingly depart from the experimentally determined values, as concentration of water increases, likely due to limitations of MD simulation approach. Based on the analysis of trajectories of diffusing water molecules resulting from MD, the role played by the different types of self- and cross-HB established in the system in determining the value of mutual binary diffusion coefficient has also been elucidated. The analysis evolution of H-bond lifetimes as it emerges from MD, provides a convincing qualitative picture of the diffusion process of water molecules in the PEI matrix.

## Data Availability

The data presented in this study and not reported in tables are available on request from the corresponding authors.

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
