# Peer review of "A Molecular Interpretation of the Dynamics of Diffusive Mass Transport of Water within a Glassy Polyetherimide"

_ijms, 2021, doi:10.3390/ijms22062908_

Round 1
Reviewer 1 Report
Correa et al. use a combined experimental and theoretical analyses to study the diffusion mechanism of water in polyetherimide. The manuscript is well written and the conclusions are supported by the data presented. I have only minor comments:
In eq. (1): if rho has the unit kg/m³ and u the unit m/s, I wonder whether n should be referred to as “mas flux density” rather than mass flux?
Has the OPLS force field been validated for this kind of system? If yes, please provide some references.
Can the authors elaborate a little bit on the type of porous structure that is formed in the fast-quenching procedure? Is it possible to provide something like a pore-size distribution of the resulting material? How dynamic is this polymer structure?
Author Response
We thank the reviewer for the comments. We included point by point answers in the attached file.
Reviewer 1
Correa et al. use a combined experimental and theoretical analyses to study the diffusion mechanism of water in polyetherimide. The manuscript is well written and the conclusions are supported by the data presented. I have only minor comments:
In eq. (1): if rho has the unit kg/m³ and u the unit m/s, I wonder whether n should be referred to as“mas flux density” rather than mass flux?
A1. We have adopted the nomenclature that, in general, is used in transport phenomena. In fact, the term ‘flux’ is defined as the rate of flow of a property (in this case mass) per unit area. The dimensions of a ‘flux’ are, then, [quantity]·[time]−1·[area]−1 (whereas the dimensions of a ‘flow’ are [quantity]·[time]−1).
Has the OPLS force field been validated for this kind of system? If yes, please provide some references.
A2. We thank the reviewer for his/her comment. As the reviewer pointed out, we used the OPLS force-field for the PEI. Our choice is based on our previous work [de Nicola et al., “Local Structure and Dynamics of Water Absorbed in Poly(Ether Imide).” Ref. [10] of the submitted manuscript] in which we presented a detailed investigation, experimental and computational, of the hydrogen bond (HB) formation in the same PEI/water system. In particular, the OPLS force-field has been tested to correctly reproduce the X-ray scattering profile of the PEI bulk, the mass density of PEI bulk and the relative amount of HB calculated in the PEI/water mixture (ad different content of water)compared with FT-IR measurements. In the following figure, the X-Ray scattering profiles of PEI bulk, from MD simulations and experimental, are compared. The figure is redrawn from the reference 10 of the submitted manuscript.
[Figure from attached file]
Comparison of calculated (red open circle) and experimental (blue line) X-Ray scattering profiles of PEI at 296 K. The data are redrawn from the reference 10 of submitted manuscript.
According to the suggestion of the reviewer, we modified the section 3.1.1 Polymer model to include more details about the model. In the following, the modified sentences have been reported in the revised manuscript.
“The atomistic model of PEI used in the present work has been taken from the ref. 10. The model is based on the OPLS-AA force-field 21-23 24 from which bonded and non-bonded interaction parameters were taken. As shown in the ref. 10, the employed model is able to correctly reproduce the X-rays cattering pattern of the polymer bulk, the mass density of PEI, and the relative amount of hydrogen bonds (compared with the experimental FTIR measurements) in the system PEI/water. ”
Can the authors elaborate a little bit on the type of porous structure that is formed in the fast-quenching procedure? Is it possible to provide something like a pore-size distribution of the resulting material? How dynamic is this polymer structure?
A3. We thank the reviewer for the interesting suggestion. We understand the point of the reviewer and its scientific relevance. In fact, we are currently working on that to better understand the morphology of the polymer matrix before, during and after the quenching process. However, that aspect will be covered in a different paper. We are sorry for that, but we do not have preliminary results about pore distributions to show. At the moment, we are developing a suitable algorithm to dentify and analyze the pore distributions in the polymer matrix. About the dynamic of the polymer structure, we discussed that point in our previous paper,where we validated the polymer model (reference 10 of the submitted manuscript). In particular, were ported the root mean square displacement (RMSD) of the center of mass (c.o.m.) of PEI chains to show that during the MD-SCF simulations each chain’ c.o.m. is covers a distance that is, at least, its size (i.e. the gyration radius Rg). Moreover, we also investigated the time needed to sample differentin dependent conformation of PEI chains. To this aim, the end-to-end distance autocorrelation function are calculated forMD-SCF simulations at different resolution degree (grid size l). Inparticular, we have found that a very short relaxation time, going from 2 to5 ns, is obtained. In the figure reported below, the RMSD and the end-to-end distance autocorrelation function are redraw from the reference 10 of the submitted manuscript.(A) Time behaviour of displacement of c.o.m. of PEI chains in unit of Rg , calculated for the MD-SCF simulation having grid size equal to Rg (2.9 nm). (B) Time autocorrelation function of the end-to-end vector for MD-SCF simulations at different grid size l.

Reviewer 2 Report
The paper “A molecular interpretation of the dynamics of diffusive mass transport of water within a glassy polyetherimide” employs experimental and theoretical methods to determine the diffusion of water in PEI. It is an interesting work, in which the authors have investigated many aspects of this issue and I would suggest its publication, if some minor points were made clear.
It seems that this work is an extension of previous works, that now presents different results. However, I noticed that Fig. 1 and Fig. 2 are exactly the same as in Ref. [10]. Is it essential to use the same Figures from your previous work?
I would also argue on the simulation method used here. In line 338 you refer to an SCP method. Did you mean the SPC (simple-point-charge) model? Reference [26] is probably wrong. Please check.
The SPC method has been widely used in water simulations in the past. However, it is not known for its accuracy. Is there a reason for using it?
There are many references that employ different models, such as the SPC-E, TIP3P, TIP4P etc. (see, for example relevant works from authors like Jorgensen, Berendsen, Abascal, Sofos etc). Please comment on the usage of SPC.
On the conclusions, the authors note that: “However, the predictions obtained from the theoretical analysis provide values of mutual diffusivity that increasingly depart from the experimentally determined values, as concentration of water increases, likely due to limitations of MD simulation approach.” Could it be because the use of the SPC model? If not, please specify in a small statement.
Author Response
We thank the reviewer for the comments. We included point by point answers in the attached file.
Reviewer 2
Comments and Suggestions for Authors
The paper “A molecular interpretation of the dynamics of diffusive mass transport of water within aglassy polyetherimide” employs experimental and theoretical methods to determine the diffusion ofwater in PEI. It is an interesting work, in which the authors have investigated many aspects of thisissue and I would suggest its publication, if some minor points were made clear.
It seems that this work is an extension of previous works, that now presents different results.However, I noticed that Fig. 1 and Fig. 2 are exactly the same as in Ref. [10]. Is it essential to uset he same Figures from your previous work?
A1.Actually, the previous work (Ref. [10]) was focused on the same system (PEI/H2O) but analysed conditions of equilibrium thermodynamics without dealing with mass transport properties.The present work is, instead, focused on the water diffusion mechanism. Section ‘2.1 Relevant results on equilibrium thermodynamics of PEI-water system’ has been written with the aim to provide a brief re-cup on the findings of the previous work considered to be relevant for the present contribution.
In particular, figure 1 is important to illustrate the types of hydrogen bonding established in the system (that are the same both at equilibrium conditions and in the course of the diffusion process)identifying the so-called first-shell and second-shell water molecules. In fact, in the MD section,these two concepts are recalled and used in the description of the diffusion process.
In addition, figure 2 has been reproposed since it clearly demonstrates how the macroscopicthermodynamic model (NETGP-NRHB) is consistent with the findings of vibrational spectroscopy analysis and of MD in the investigation of the system at equilibrium. This is a relevant point since,in the present contribution, the NETGP-NRHB model is used to calculate the ‘thermodynamic contribution’ that appears in the expression of the mutual diffusion coefficient.
I would also argue on the simulation method used here. In line 338 you refer to an SCP method. Didyou mean the SPC (simple-point-charge) model? Reference [26] is probably wrong. Please check.The SPC method has been widely used in water simulations in the past. However, it is not known for its accuracy. Is there a reason for using it?
There are many references that employ different models, such as the SPC-E, TIP3P, TIP4P etc. (see,for example relevant works from authors like Jorgensen, Berendsen, Abascal, Sofos etc). Please comment on the usage of SPC.
On the conclusions, the authors note that: “However, the predictions obtained from the theoretical analysis provide values of mutual diffusivity that increasingly depart from the experimentally determined values, as concentration of water increases, likely due to limitations of MD simulation approach.” Could it be because the use of the SPC model? If not, please specify in a small statement.
A2. We thank the reviewer for the comment. The reviewer is right, the acronym SPC indicates thesimple-point-charge model. Besides, after a double-check, we confirm that the reference 26 of thesubmitted manuscript is correct.
As the reviewer pointed out, due to the relevance of water to so many fields, several models of water (with growing complexity and accuracy of thermodynamic properties) have been published. For that reason, atomistic model of water can be very simple, as the two-dimensional model proposed by Silverstein [K. A. T. Silverstein, A. D. J. Haymet, and K. A. Dill, J. Am. Chem.Soc. 120, 3166 1998.], or very complex including additional interaction site, dipole, explicit polarizability or smeared charges [C. J. Burnham and S. S. Xantheas, J. Chem. Phys. 116, 51152002, J. Jeon, A. E. Lefohn, and G. A. Voth, J. Chem. Phys. 118, 7504 2003]. On the other hand, a higher complexity of the model has direct impact on the computational cost, which can unbalance the cost/benefit ratio (in terms of accuracy of relevant properties) limiting the employ of those complex models.
We employed the SPC model because has the best cost/benefit ratio in terms of reproduction of the self-diffusion coefficient of liquid water in bulk (D(water)), and the computational cost to employ it. In the table reported below, a comparison of the D(water) calculated from MD simulations at 298 and 301 K, for different water models, is shown. As can be seen, all models overestimate theexperimental D of water.3,4 In particular, the SPC overestimates the D(water) of a factor ~1.7, while afactor ~2.4 if found for the model TIP3P (is the worst case). Considering the computational cost toemploy a water model, and the reproduction of D(water),the best choice is the SPC. In addition, the lessexpensive SPC model has revealed to well-reproduce the relative amount of hydrogen bonds in the mixtureof PEI/water.Table.
[please check Table from attached file]
Wu, Y.; Tepper, L. H.; Voth, G. A.; J. Chem. Phys. 2006, 124, 024503.2. Mark, P.; Nilsson, L.; J. Phys. Chem. A, 2001, 105, 9954-9960.3. K. Krynicki, C. D. Green, and D. W. Sawyer, Faraday Discuss. Chem. Soc. 66, 199 (1978).4. Mills, R. J. Phys. Chem. 1973, 77, 685.
As we reported in the revised manuscript and pointed out from the reviewer, we observed a discrepancy of the mutual diffusivity that increasingly depart from the experimentally determined values, as the concentration of water increases (see Figure 8 of the revised manuscript). About that point, we should underline that the simulated systems are not simply water bulk systems. In our case, we simulated systems composed of a polymer matrix of PEI, and a certain number of water molecules (at different concentrations)inserted in it. As can be seen from Figure 8, as the content of water increases, the discrepancy of the mutualdiffusivity of water increases too. In particular, at lower content of water, the interactions between PEI andwater dominate over the water-water interactions. In that case, our model well reproduces the diffusivity(lower discrepancy). Differently, when the water content increases, and hence approaching the bulk densityof the water where the water-water interactions dominate, the effect of the overestimation of D(water) calculatedfrom MD simulations becomes larger, and the discrepancy increases. This is not surprisingly since all watermodels overestimate (more or less) the experimental diffusion coefficient of water in bulk, that discrepancycould occur, in principle, with any model. For that reason, we cannot strictly attributes that discrepancy to alimit of the MD technique, but in our opinion is more likely to depend on the model of water.
[Figure 8 from the revised manuscript.]

Reviewer 3 Report
- Time-resolved difference spectra that were analyzed in Eq. (16) should be presented. What is "analytical band" for A(t)?
- Time variable is missing in Eq. (16).
Author Response
Reviewer 3
Comments and Suggestions for Authors
1.Time-resolved difference spectra that were analyzed in Eq. (16) should be presented. What is "analytical band" for A(t)?
A1. We thank the reviewer for the comment. In the revised manuscript, the time-resolved differencespectra have been added as an inset of Figure 3. The analytical band for A(t) is the ν(OH) water band in the 3800 – 3250 cm-1 range.
In the revised manuscript, at pag. 11, lines 395 – 396, the caption of Fig. 3 has been updated and thefollowing sentence has been added in the text to make this point clearer.
“The inset displays the ν(OH) water band at increasing sorption times.”
[please check Figure 3 from attached file]
Figure 3. Fick’s plot [A(t)/A(∞) vs √t] for the sorption test at p/p0 = 0.6 at T=303.15 K. The inset displays the time-evolution of the analytical band.
2.Time variable is missing in Eq. (16).A2. We thank the reviewer for the comment. The equation (16) has been amended.
